# *Saccharomyces cerevisiae* as a Toolkit for COP9 Signalosome Research

**DOI:** 10.3390/biom11040497

**Published:** 2021-03-25

**Authors:** Dana Harshuk-Shabso, Noam Castel, Ran Israeli, Sheri Harari, Elah Pick

**Affiliations:** 1Department of Human Biology, Faculty of Natural Sciences, University of Haifa, Haifa 31905, Israel; harshukdana@gmail.com; 2Department of Evolutionary and Environmental Biology, Faculty of Natural Sciences, University of Haifa, Haifa 31905, Israel; noamcastel@gmail.com; 3Department of Biology and Environment, Faculty of Natural Sciences, University of Haifa at Oranim, Tivon 36006, Israel; ranisraeli1988@gmail.com (R.I.); sherryperets@gmail.com (S.H.)

**Keywords:** COP9 signalosome, NEDD8, Rub1, CSN5/Jab1, CSN5i-3, cullin RING E3 ligase

## Abstract

The COP9 signalosome (CSN) is a highly conserved eukaryotic multi-subunit enzyme, regulating cullin RING ligase activities and accordingly, substrate ubiquitination and degradation. We showed that the CSN complex of *Saccharomyces cerevisiae* that is deviated in subunit composition and in sequence homology harbors a highly conserved cullin deneddylase enzymatic core complex. We took advantage of the non-essentiality of the *S. cerevisiae* CSN-NEDD8/Rub1 axis, together with the enzyme-substrate cross-species activity, to develop a sensitive fluorescence readout assay, suitable for biochemical assessment of cullin deneddylation by CSNs from various origins. We also demonstrated that the yeast catalytic subunit, CSN5/Jab1, is targeted by an inhibitor that was selected for the human orthologue. Treatment of yeast by the inhibitor led to the accumulation of neddylated cullins and the formation of reactive oxygen species. Overall, our data revealed *S. cerevisiae* as a general platform that can be used for studies of CSN deneddylation and for testing the efficacy of selected CSN inhibitors.

## 1. Introduction

Cells, tissues, and organismal health depend on a complex network of homeostasis between protein synthesis, functioning, and degradation, which are collectively referred to as “proteostasis”. When proteostasis breaks down, diseases such as cancer, inflammation, or neurodegeneration can appear [1]. Indeed, various components of proteostasis are attractive targets for therapeutic intervention [2]. Maintenance of proteostasis requires timely degradation of proteins, which are damaged by incorrect translation, and improper folding during synthesis, genetic mutations, or are no longer required due to changes in environmental ques or cellular needs [3]. Two major proteolytic machineries, the ubiquitin proteasome system (UPS) and the lysosomal autophagy, are essential for maintaining cell health [4,5,6].

Rates of degradation by the UPS have generally been determined by the modification of substrates by ubiquitin (Ub) [7,8]. The ubiquitination of UPS substrates requires an enzymatic cascade, starting by E1 (Ub-activating enzyme), which forms a thioester with Ub following ATP consumption. Next, an E2 (Ub-conjugating enzyme) takes over Ub from the E1, forming again thioester and transferring the Ub to a specific substrate, selected by E3 (Ub ligase), which can further form a poly-Ub chain [9,10,11]. Substrates conjugated primarily with Lys48-linked poly-Ub chains are transferred to the 26S proteasome complex for degradation [6,12,13]. The 26S proteasome is symmetric and composed of two sub-complexes: The 20S catalytic core particle is located in the center and two 19S regulatory particles surrounding the 20S on both sides. The distal part of the 19S particle is referred to as the 19S lid subcomplex [14]. The 19S lid is composed of eight endemic non-ATPase subunits (Rpn3, Rpn5–9, and Rpn11–12) and DSS1/Sem1, which is found in additional complexes [14,15]. Interestingly, the 19S lid is a close homologue of two other complexes: The COP9 signalosome (CSN) and the eukaryotic translation initiation factor 3 (eIF3), which together have been termed as proteasome lid, CSN, and eIF3 (PCI) complexes [14,15,16]. PCI complexes contain a pair of Mpr-1–PAD1–N-terminal (MPN) domain-containing subunits, sometimes bearing an MPN^+^ metal-binding motif, also known as JAB1-MPN-MOV34 (JAMM) [17,18]. The biochemical role ascribed to the 19S lid is deubiquitination activity of the MPN^+^/JAMM metalloenzyme, Rpn11 [19,20]. Similarly, the CSN complex also harbors metalloprotease activity ascribed to CSN5/Jab (aka Csn5 in *Saccharomyces cerevisiae*), a direct paralogue of Rpn11 [21]. The great similarity between CSN5 and Rpn11 leads to the fact that their substrates are also paralogous—various Ub-linked proteasome targets for Rpn11; and NEDD8 (aka Rub1 in *Saccharomyces cerevisiae*) conjugated proteins, namely cullins, for Csn5 [22]. Each cullin serves as a platform for building a modular array of multi-subunit E3 enzymes, known as cullin-RING ligases (CRLs). CRLs are required for the ubiquitination of a wide variety of 26S proteasome targets, and the neddylation of cullins activates all of them (Figure 1A) [23,24].

The CSN is a highly conserved complex that was first purified from plants as a repressor of light-dependent growth patterns during darkness [27,28]; and from mammalian tissues as a signaling regulator [29]. CSN complexes are essential for the vitality of multicellular organisms, and their loss-of-function mutants display critical pleiotropic phenotypes [30,31,32]. CSN essentiality has been demonstrated in *Arabidopsis thaliana*, where loss of any of the subunits proved lethal in the development of seedlings. Similarly, in mice, knockdown of CSN subunits is fatal in the early embryonic stage [33]. Although critical in multicellular organisms, the loss of CSN subunits in unicellular organisms does not always lead to inviability; examples are seen in various fungal species, including *Aspergillus nidulans*, *Neurospora crassa, Schizosaccharomyces pombe,* or *Saccharomyces cerevisiae* [34,35]. Cullin deneddylation is performed only when CSN5 resides in an intact CSN, attached to the CRL [16,17,18,21]. In human, CSN–CRL interactions lead to a line of conformational changes within the CSN complex, starting by attachment of CSN4 to the cullin subunit, triggering rearrangements in CSN6, and finally priming a glutamate 104 residue in CSN5 for deneddylation [36,37]. Recently, CSN5i-3, a specific and orally available inhibitor of the human CSN5, was selected via a high-throughput screen for small molecule inhibitors [26]. CSN5i-3 interferes with the UPS by blocking the deneddylation of a subset of CRLs and therefore suggests them as druggable targets.

The CSN complex also regulates CRLs in a non-catalytic manner through steric clashes that inhibit the recognition of substrates for ubiquitination by CRL substrate receptors, and by preventing the association of CRLs with Ub E2 enzymes, which are required to ubiquitinate these substrates (Figure 1A, right) [36,38]. When it comes to complex integrity, most CSN subunits normally exist in a stable complex, whereas CSN5 is unique in its distribution both as part of a CSN holocomplex and in free form (CSN5-f) [39,40]. CSN5-f is a pro-proliferative factor [41] that interacts with the cyclin-dependent kinase Cdk2 [42], mediates nuclear export followed by degradation of the CDK inhibitor p27, and interacts with activator protein 1 (AP-1) to mediate proliferation [43].

Not all of these functions of CSN5-f/CSN are conserved across phyla. Indeed, the most diverged CSN complex was found and characterized in the budding yeast *S. cerevisiae* [44,45,46,47]. This complex excludes orthologues for CSN3 and CSN8 [48]. The orthologue of CSN6 (aka Csi1) is diverged and lacks the characteristic amino acid sequence for MPN domain [44]; the core subunit CSN4 is absent and replaced by the paralogous subunit within the 19S lid, Rpn5 [45]. In this manuscript, we describe a sensitive fluorescence readout assay, suitable for biochemical assessment of cullin deneddylation by CSNs from distinct organisms, such as *Homo sapiens* and *S. cerevisiae*. We also exhibit that the *S. cerevisiae* Csn5 is inhibited by CSN5i-3, an inhibitor that targets the human orthologue. Altogether, we demonstrate that despite the diverged composition, the architecture and enzymatic activity of the CSN cullin deneddylase are highly conserved from yeast to human.

## 2. Materials and Methods

### 2.1. Strains and Plasmids

All plasmids were isolated from *Escherichia coli* DH5 alpha cells using an alkaline lysis plasmid preparation method. Plasmids were either used in previous research or cloned for this study as indicated in Appendix A. All single deletion *S. cerevisiae* mutants were purchased or obtained as specified in Appendix A. The double mutant of *Δcsn5/Δrub1* was formed by mating and sporulation between the single mutants.

### 2.2. Plasmid Cloning

Cdc53/yCul1 truncation mutants were performed through PCR amplification of *CDC53* from genomic DNA using primers (Appendix A), followed by DNA sequencing. The amplified DNA was subcloned by recombination into the Yeplac181 plasmid including the alcohol dehydrogenase1 (*ADH1*) promoter, Regulator of G-Protein Signaling 9 (RGS), and 8His tags (EP153), generating plasmids EP172, EP237, EP238, and EP247.

Subcloning of Rub1 was performed through PCR amplification using specific primers (Appendix A). The PCR product was cloned by recombination downstream to *ADH1* promoter and a Green fluorescent protein (GFP) tag, generating plasmid EP233.

### 2.3. Growth Conditions

The various yeast strains were grown in a rich YPD medium (yeast extract 1%, peptone 1%, dextrose 2%), unless specified otherwise. Plasmids were maintained by culturing the plasmid-containing strains in a selective synthetic complete medium based on a yeast nitrogen base (YNB) supplemented with ammonium sulfate, in which a complete mixture of amino acids supplements each of the commonly encountered auxotrophies. The various inhibitors (cycloheximide, MG132, and CSN5i-3) were dissolved in dimethyl sulfoxide (DMSO) before being added directly to the growth medium as indicated.

### 2.4. Native Protein Extraction

Cells were cultivated for 24–48 h then washed twice with DDW and re-suspended in two volumes of Y-cell lysis buffer (Sigma Aldrich, Rehovot, Israel). For extraction, 0.5-mm sized zirconia beads were added followed by vortexing the samples for 1 min, 10 times, with 1 min intervals on ice. Native lysates were clarified by centrifugation at 16,000× *g* for 10 min at 4 °C and transferred to a clean tube.

### 2.5. Calmodulin-Based Affinity Purification of the CSN

*S. cerevisiae* CSN purification was performed through the affinity tagged version of Csn10, harboring a Calmodulin Binding Peptide (CBP) at the C terminal as previously described [45]. The elution was used for immunoblotting or subjected for CSN activity tests.

### 2.6. Immunoblotting

Grown cultures were harvested in trichloroacetic acid (TCA) as previously described [45]. Following the extractions, samples were resolved by SDS-PAGE and transferred to a nitrocellulose membrane for immunoblotting as presented in the figure legends. Experiments were repeated at least three times and a representative image is shown.

### 2.7. Purification of the Substrate by Ni-NTA Affinity Chromatography

Ni-NTA beads were washed twice by Y-cell lysis buffer (Sigma-Aldrich, Rehovot, Israel), then washed twice with Tris-buffered saline (20 mM Tris, pH 7.5. 150 mM NaCl) complemented by 0.1% Tween 20 before added to native protein lysate of cells co-expressing 8His-yCul1 and GFP-Rub1. Samples were placed on a rotator overnight at 4 °C, and then centrifuged in 3000× *g* for a minute. Elutions were used for immunoblotting, fluorescent readout in a microplate reader (Biotek Synergy HT, Winooski, VT, USA) (excitation: 485 nm/20, emission: 528 nm/20), or subjected for CSN activity assay.

### 2.8. Fluorescence Microscopy

Early logarithmic phase cultures (OD_600_ nm = 0.8–1.0) were centrifuged and washed once with phosphate buffer saline (PBS) before the addition of dyes (BODIPY or FM4-64). Differential interference contrast (DIC) and fluorescence of cells were evaluated by a microscope (NIKON Eclipse E600, Tokyo, Japan) and photographed according to Sinha et al., 2020 [48].

### 2.9. Ergosterol Extraction and Analysis

The evaluation of ergosterol quantity in yeast cells was performed as previously described [48].

### 2.10. Endogenous Oxidative Stress Measurements

Endogenous reactive oxygen species (ROS) was measured by fluorescence following treatment with 2′,7′-dichlorofluorescein diacetate (DCFDA) (Sigma-Aldrich, Rehovot, Israel), which was directly added the cultures as described [49].

### 2.11. CSN Activity Assay

Total extracts of yeast cells expressing 8His-Cul1-GFP-Rub1 substrate were mixed with either human CSN complex (Protein Center, Ramat Yohanan, Israel https://proteasome.net/; accessed on 20 December 2018) or with total cell extracts of wildtype yeast cells (harboring the endogenous yeast CSN complex). CSN activity was performed on a rotator for 30 min at room temperature. At the next step the substrate was affinity purified for 45 min on Ni-NTA resin at 4 °C followed by three washes with TBST × 0.5 and elution from the resin with 250 mM imidazole. The fluorescence readout of eluted proteins was quantified by a plate reader (excitation: 485 nm/20, emission: 528 nm/20). Fluorescent values were compared with a CSN-free control experiment. CSN activity was calculated as the ratio between the residual fluorescence in the CSN-containing assay divided by the control fluorescence.

## 3. Results and Discussion

Although Rub1 is not required for the *S. cerevisiae* viability [50], the neddylation pathway in this organism is highly conserved, including a viable replacement of the *S. cerevisiae* Rub1 with the mammalian NEDD8 that can modify Cdc53/yCul1 [51]. Molecular modeling suggests evolutionary conservation of the binding sites for either ATP or NEDD8/Rub1 in the E1 enzyme (NAE1) of human and yeast, also reflected by superimposition of MLN4924 (Pevonedistat), a selective inhibitor for NAE1 (Figure 1B; Appendix A). Unlike the conserved neddylation pattern, CSN composition is more diverged (Table 1; Appendix A) [37,52,53,54]. Indeed, the *S. cerevisiae* CSN subunits cannot be replaced by their orthologues [45]. Yet, considering cullin deneddylation, *S. cerevisiae* and human CSN complexes demonstrate cross-species enzymatic activity: Purified CSN complexes from each origin can hydrolyze NEDD/Rub1 from CRLs of the other organism in total cells extracts [44,55]. According to the previously described orientation (Figure 1C, left [26]), we predicted the binding-conformation of the small molecule ligand, CSN5i-3, to the MPN+/JAMM motif of the *S. cerevisiae* Csn5 (Figure 1C, right). The superimposition confirmed high conservation of the CSN MPN+/JAMM motif across phyla. Indeed, the predicted structure of *S. cerevisiae* Csn5 shows a high similarity to four published human CSN5 structures (Appendix A).

### 3.1. S. cerevisiae Harbors a Highly Conserved Enzymatic Core Complex

The C-terminal domain of CSN6 has an indispensable role in maintaining the full integrity of the CSN complex, even in a truncated mutant lacking the MPN^−^ domain [44]. These data were confirmed by Lingaraju et al., 2014 [37], who resolved the crystal structure of the human CSN in a resolution of 2.8 Å. The structure elucidated a horseshoe-shaped molecular architecture, formed by the N-terminal of PCI subunits, surrounding a helical bundle built from C-terminal of all subunits, with both MPN domains of CSN5 and CSN6 located above it. Unlike the C-terminal of CSN6, which interacts with all other subunits, CSN5 creates fewer interactions, which could indicate on a more dynamic nature [39,40]. In a previous study, we identified the *S. cerevisiae* Csi1 as a diverged CSN subunit, displaying homology with the C-terminal of the canonical Csn6 (a.k.a. S6CD), but lacking the MPN domain. To confirm that Csi1 also interacts with various subunits and maintains the CSN architecture, we approached the protein-fragment complementation assay (PCA) [61]. In this assay, the integrity of *S. cerevisiae* cytosine deaminase (FCY1) fragments, each of which is attached to a different protein, was used as a reporter that allowed for a survival assay by deaminating cytosine to the essential uracil. The results revealed direct interactions between Csi1 and three CSN subunits: Csn5/CSN5, Csn9/CSN7, and Rpn5/CSN4 (Appendix A). To complete the picture, a calmodulin-based affinity pulldown of a chromosomal integrated Csn10/CSN2 TAP-tagged at the C-terminal (CBP followed by two immunoglobulin G-binding domains of protein A) revealed an internal interaction between Csn10/CSN2 and both CSN9/Csn7 and Rpn5/CSN4, even in the absence of Csn5 (Figure 2A,B). The above suggests that similarly to CSN6 in canonical CSN complexes, the yeast Csi1 is a central element in the complex integrity, while Csn5 shows more freedom.

Interestingly, molecular modeling of Csi1 on available structures of the human orthologue of CSN6 by Swiss-Model suggested an N-terminal MPN structural fold. Since Swiss-Model is highly reliant on the correct template provided for modeling, we ran structure prediction on Csi1 using Phyre^2^. Interestingly, using this modeling tool, the closest model for Csi1 was found as PDB ID 5A5T of the human eIF3F (98.0% confidence, 13% sequence identity), one of the two MPN subunits of the eIF3, which is in fact a paralogue of CSN6/Csi1. CSN6 appeared second on the Phyre^2^ list, with 97.5% confidence and 15% sequence identity. We used CSN6 (Figure 2C) and eIF3F (Appendix A) as templates to predict Csi1 structure. The superimposition of Csi1 (dark blue) with either CSN6 or eIF3F (light blue) revealed a possible MPN fold at the N-terminal areas of the *S. cerevisiae* Csi1. This estimated structure could not be predicted in the past, due to poor structural data [44]. The above suggests that the MPN dimer (CSN5–CSN6 dimer) could be highly conserved. Given these considerations, we approached a modeling method to confirm the high conservation of the deneddylase core complex, composed of Csn4, Csn6, and Csn5 [37] (Figure 2D and Appendix A). Overall, this implies that the CSN deneddylase core complex may be more conserved than estimated. Nevertheless, the presence homology modeling is partially based on low coverage (Table 1; Appendix A), hence, additional structural studies are required.

### 3.2. Utilizing the S. cerevisiae Neddylation Pathway to Develop a Fluorogenic CSN Substrate for In Vitro Studies

The conserved CSN deneddylase core complex (Figure 1C and Figure 2D) and the cross-species enzymatic activity [44,55] have led to the speculation that yeast can be approached instrumentally as a tool for developing a general substrate for CSN activity assay. The *S. cerevisiae* genome encodes for three cullin proteins (yCul1/Cdc53, Cul3, and Cul4/Rtt101), each of which serves as a building block for the assembly of multi-subunit CRLs [62,63,64,65]. The yCul1-based CRL is an archetype of the family, also known as the Skp1 Cul1 F-box (SCF) [64]. We selected yCul1 to evaluate if the endogenous neddylation cascade tolerates the modification of its overexpressed version by a co-expressed GFP-Rub1, in a double mutant yeast strain, lacking CSN activity and endogenous Rub1 (Figure 3A). We speculated that such a modified cullin could fit as a substrate for CSN from various origins and would be valuable for a quantitative understanding of CSN biochemical properties. Because yCul1 is a large protein, essential for yeast viability, we wanted to select a protein size with a minimal effect on yeast vitality. With this in mind, we designed a number of yCul1 truncation mutants, all of them included the neddylation site, Lys 760, located at the C-terminal domain of yCul1 [66], yet shorter in the N-terminal substrate recognition domain. For affinity purification purposes, each of the mutants included the N-terminal 8His tag for the binding of Ni-NTA affinity resin and withstood extensive washing to remove non-specifically bound proteins (Figure 3B, top). Both the ability to be modified by endogenous Rub1 and recognition by the CSN were determined for each truncated mutant by immunoblotting in wildtype *Δcsn5* or *Δrub1* mutant strains, as summarized in the table (Figure 3B, right).

Each of the truncation mutants was ectopically expressed in a multi-copy plasmid under the control of *ADH1* promoter. With the distinction of C192, all of them were expressed in the cells, and sizes larger than C377 were found neddylated by endogenous Rub1 in *Δcsn5* cells (data not shown). Since yCul1 is a key cell cycle regulator [67], we evaluated if expression of the truncated mutants interferes with growth or morphology. Indeed, ectopic expression of C293 led to very small-sized yeast colonies that could not be cultivated in broth medium or transferred to fresh plates from the transformation plates. Examination of cell morphology by microscope suggested severe cell cycle defects (Figure 3B, bottom). Although the truncation mutants did not provide a practical advantage to the assay, they can still be useful, especially given the dominant negative activity of C293. This mutant lacks the N-terminal arm of yCul1 that is involved in the selection of SCF substrates, and could thus be suggested as an original way to identify additional SCF substrates that have not yet been identified. Note that both FL815 and C548 showed a similar neddylation ratio upon logarithmic growth (Appendix A). The C548 truncated mutant was generated according to Scott et al., 2010 [66], which produced a similar mutant (a.k.a. “C+”), for in vitro yCul1-Skp1 binding assays. Their results showed that C548/C+ cannot interact with the adaptor protein Skp1. Knowing that SCF assembly (i.e., the assembly of yCul1 with Skp1 and Rbx1) is required for vitality, it was surprising that overexpression of this mutant did not interfere with the SCF assembly by causing a cell cycle arrest. Contrary to expectations, the C548/C+ even presented effective yCul1 neddylation in *Δcsn5* (Appendix A, top). We assumed that at least in vivo, the C548/C+ is sufficient for the Skp1–yCul1 interaction. This possibility is supported by the finding of Kurz et al., 2005, which showed that Skp1 is necessary, to a certain extent, for neddylation [68]. We speculated that dissimilarities between in vitro and in vivo data are due to other protein–protein interactions, either through other complex counterparts or by assisting chaperons. However, since C548 showed slight abnormal cell morphology (Figure 2B) and had no advantage compared with FL815, we eventually approached the full size of yCul1, FL815, to standardize the cullin deneddylation assay.

To confirm that the expression of GFP-Rub1 improves the neddylation properties of the co-expressed 8His-yCul1, plasmids were co-transformed into *Δcsn5Δrub1* cells. Immunoblotting for anti-6His (to recognize the 8His-yCul1), anti-GFP, or anti-Rub1 (to recognize GFP-Rub1) revealed a protein at the same marker size as 8His-yCul1-Rub1-GFP, suggesting that the intrinsic *S. cerevisiae* neddylation pathway is sufficient for the modification of overexpressed 8His-yCul1 by the co-expressed N-terminally GFP-tagged version of the modifier (Figure 3C). Similarly, the affinity purification of 8His-yCul1 via Ni-NTA followed by imidazole elution led to a high fluorescence readout compared to imidazole containing buffer, or to mock purification using naïve wildtype cell extracts (Figure 3D). In order to develop a readout assay, in which the fluorogenic product (i.e., GFP-Rub1) is efficiently hydrolyzed from 8His-yCul1, we repeated the purification without eluting the substrate from the resin. Accordingly, *S. cerevisiae* CSN was purified from wildtype cells, expressing a chromosomal Csn10-TAP and kept attached to the calmodulin resin. Deneddylation was carried out by mixing resin-attached enzymes with resin-attached 8His-yCul1-GFP-Rub1 substrates for 20 min at 30 °C. Following the assay, the beads were washed to remove the released GFP-Rub1 product, and the residual fluorescence of the substrate while still attached to the beads was measured and compared with the fluorescence of the experiment buffer, CSN alone, or the untreated 8His-yCul1-GFP-Rub1 substrate (Figure 4A).

The expected result was a decline in fluorescence of the resin-attached substrate, co-treated with the CSN. Although repeated in three independent experiments, the high background and only small differences in the relative fluorescent units (RFU) compared to controls indicated the futility of this assay as a CSN readout assay in the long term. Among all other conditions that were evaluated, we found the strategy of adding resin-free CSN to a crude extract of *Δcsn5Δrub1* cells expressing the 8His-yCul1-GFP-Rub1 substrate as the best option (Figure 4B). In this experimental setting, the 8His-yCul1-GFP-Rub1 substrate was subjected for Ni-NTA purification only following the experiment. CSN activity was calculated as the ratio between the fluorescence of a fixed amounts of substrate exposed to the CSN, or to buffer used as a control. Accordingly, substrates that had been pretreated with human CSNs were compared to the buffer-treated substrates, and crude extracts of *S. cerevisiae* wildtype cells (containing endogenous CSNs) were compared to crude extracts of *S. cerevisiae Δcsn5* mutant cells lacking CSN activity (Figure 4B). The low fluorescence of CSN-containing experiments compared to the controls suggested a highly detectable cullin deneddylase activity of CSN complexes from diverged organisms. In conclusion, we were able to develop a general fluorescent substrate, available for in vitro CSN deneddylation activity. Given that *S. cerevisiae* 8His-yCul1-GFP-Rub1 is an effective substrate for complexes from the most distinct organisms, it may serve as a universal substrate for evaluating the activity of CSN found in most, if not all, eukaryotes.

### 3.3. CSN5i-3 Targets CSN5/Csn5 from Diverged Eukaryotic Sources

The predicted docking position of CSN5i-3 indicates that the behavior of this small molecule in the human or *S. cerevisiae* CSN active site is alike at the atomic level (Figure 1C); thus, CSN5i-3 might be useful for CSN studies in *S. cerevisiae*. To evaluate this supposition, we approached *∆pdr5* (pleiotropic drug resistant), a mutant yeast strain that allows resistance to an array of drugs to be overcome, and therefore promoting the entry of CSN5i-3 into cells. As an initial proof of concept, the concentration that most effectively inhibits CSN activity in *∆pdr5* cells was selected. Accordingly, cells were grown to the early logarithmic phase in a glucose-rich YPD media, before the addition of CSN5i-3 for the specified times (Figure 5A).

Notably, 100 µM of CSN5i-3 was compared with the highest concentrations that was used for Csn5 targeting in human cells [26]. The effect of CSN5i-3 was studied by evaluating if the treatment led to a higher neddylation status. The immunoblots obtained show that CSN5i-3 has an effect that decreases overtime, only at concentrations higher than 60 µm (Figure 5A). Effective inhibition of yCul1 deneddylation was seen upon incubation with 100 µM of the inhibitor for 2 h. To confirm that the change in yCul1 neddylation ratio was not due to modulation in protein expression/stability, the ratio of neddylated to free yCul1 was assessed following cells treatment by cycloheximide (CHX) to inhibit protein expression, or MG132 to inhibit proteasomal degradation. Immunoblots showed that CSN5i-3 causes accumulation of yCul1 neddylation (~30% difference) upon co-treatment of CHX with CSN5i-3 (Figure 5B, lane 3 vs. lane 7), altogether suggesting that CSN5i-3 had effectively inhibited the deneddylation of *S. cerevisiae* yCul1, hence, allowing the use of CSN5i-3 for studying *S. cerevisiae* CSN activity in vivo, in various environmental or metabolic conditions.

Our previous findings revealed metabolic and induced accumulation of ROS as a molecular switch-off of the neddylation cascade. Indeed, the addition of hydrogen peroxide (H_2_O_2_) led to a dramatic decrease in cullin neddylation and the accumulation of cullin-free Rub1 [49]. The modulation of yCul1 neddylation status in an oxidative environment from highly to slightly modified may indicate CSN activity during oxidative stress. To evaluate this possibility, we treated *∆pdr5* cells with H_2_O_2_ and evaluated if co-treatment with 100 µM of CSN5i-3 altered yCul1 neddylation status. Unexpectedly, unlike wildtype, cullin neddylation status in *∆pdr5* cells was not sensitive to H_2_O_2_ (Appendix A). Due to these findings, the efficacy of various concentrations of CSN5i-3 was evaluated in wildtype logarithmic cells as well. As expected, the dose of CSN5i-3 that targets Csn5 in wildtype cells and leads to elevated yCul1 neddylation status was higher (200 µM) than in the *∆pdr5* mutant (100 µM) (Figure 6A compared with Figure 5A). We used this concentration to assess if CSN activity is maintained in an oxidative environment. For this purpose, logarithmic wildtype cells were treated by H_2_O_2_ for 10 or 30 min following the addition of CSN5i-3 (200 µM). Indeed, the addition of H_2_O_2_ led to increased performance of non-neddylated yCul1 (Figure 6B, lane 2). Yet, the pretreatment with CSN5i-3 slowed down the effect of H_2_O_2_, which was reflected in a smaller amount of non-neddylated yCul1 compared to the untreated sample, implying that Csn5 is active during oxidation (Figure 6B, lane 4). The insensitivity of Csn5 activity to H_2_O_2_ is not surprising since MPN+/JAMM metalloproteases exclude cysteine at the active site [69]; however, this is the first study to show constitutive CSN activity after H_2_O_2_ oxidation. Moreover, we evaluated if CSN subunit expression is altered in growth phases characterized by accumulated ROS (i.e., the diauxic shift). Our data revealed that at least one subunit, Csn10, expresses higher in the diauxic and post-diauxic phases (9–24 h) than in the logarithmic growth phase (6 h) (Figure 6C). Overall, the above indicates that the CSN is expressed and active in an oxidative environment.

The induction of CSN expression at the oxidative phase could suggest that this enzyme is required for oxidative stress tolerance. Indeed, treatment of *∆csn5* cells with DCFDA revealed that Csn5-deficient cells produce high ROS (Figure 6D). The findings that *∆csn5* mutant cells produce high ROS may be in conflict with the previous finding that yCul1 is fully neddylated in this mutant [46], since ROS inhibits the neddylation cascade [49]. Accordingly, there should have been a population of non-neddylated cullins as well. Despite this estimation, even a treatment of *∆csn5* mutant cells with H_2_O_2_ revealed the total accumulation of neddylated yCul1, yCul3, and Rtt101, the three *S. cerevisiae* cullins (Figure 6E). The results were in line with the finding that treatment of wildtype cells with CSN5i-3 led to a high yCul1 neddylation status, even though the cells produced ROS (Figure 6F). These findings can be explained if single-enzyme removal of Rub1 is a more efficient process than the reverse process that requires a series of enzymes.

Considering that CSN activity does not regulate the turnover of *S. cerevisiae* CRLs, the results could imply that the release of Rub1 from yCul1 is required to tackle cellular oxidative states through a CRL-free mechanism. Indeed, several studies suggested that NEDD8 and Rub1 orthologues are involved in the cellular response to oxidation [49,70,71]. To evaluate if the accumulation of ROS in *∆csn5* is due to the lack of deneddylation, logarithmic wildtype cells were treated by CSN5i-3 (0, 25, 75, and 200 µM) for 2 h followed by DCFDA for one hour prior to harvest, and the percentage of fluorescent cells was calculated. Our data revealed that 25 µM of CSN5i-3 induced ROS in wildtype cells (Figure 6E). This information is surprising because no effect on yCul1 neddylation was observed after treatment with 25 μM CSN5i-3 (Figure 6A). Moreover, higher concentrations of CSN5i-3 led to higher ROS also in the *∆csn5* mutant cells, leading to the suspicion that the inhibitor has other effects on cell physiology, in addition to CSN inhibition. These aspects should be studied in details. Notably, by using CSN5i-3, we did not observe in wildtype other phenotypes of *∆csn5* such as defects in ergosterol biosynthesis or enlarged vacuoles [48] (Appendix A). Considering the generation time of *S. cerevisiae* (~2 h), we speculate that the lack of physiological phenotypes, such as alterations in lipids, was due to the short exposure of cells to the inhibitor (2 h). The short exposure might promote rapid metabolic changes in respiration or accumulation of ROS without inducing changes resulting from the synthesis or breakdown of macromolecules like lipids. These results should be further examined in the future, after prolonged treatment of cell with the drug. In conclusion, we were able to show through subunit interactions and models of homology that the CSN deneddylase center is conserved. This has been proven by the development of a general substrate (8His-yCul1-GFP-Rub1) and identification of a general inhibitor (CSN5i-3), suitable for CSN studies in diverged eukaryotes.

## 4. Concluding Remarks

In this study, the simplest eukaryotic model organism, *S. cerevisiae,* which harbors the most diverged CSN complex, was harnessed instrumentally for CSN studies. Endogenous neddylation enzymes within this organism have been used to engineer a purifiable version of SCF that is covalently attached to a fluorogenic Rub1 that successfully used as a general substrate for a CSN activity readout assay, measured by a fluorimeter. This universal, fast, and cheap assay had an advantage over other methods [72]. Future studies will include adjustment of the assay for detection of endogenous levels of CSN activity, which could further serve as an inexpensive universal method for evaluating enzyme activity even in non-model organisms in their natural habitats. Such research can be of great value, especially when considering the growing interest in environmental research, including the interplay between genome-based biology and environmental exposures, including climate change and outcomes such as global warming, drought, oxidation, salinity, pollution, or radiation.

Csn5-i3 was initially selected for inhibition of the human CSN. Considering the evolutionarily conserved MPN+/JAMM motif, it is not surprising that the inhibitor is also targeted by the yeast complex. Our results suggest that high doses of the inhibitor affect additional physiological pathways in yeast, which will need to be further deciphered. The data could only be obtained because of the non-essentiality of Csn5 in this organism. We expect these studies to be a cornerstone for selecting additional derivatives of CSN inhibitors [73], and to examine their possible side effects in the *S. cerevisiae ∆csn5* mutant, or in other fungal species that sustain non-essential CSNs.

## Figures and Tables

**Figure 1 biomolecules-11-00497-f001:**
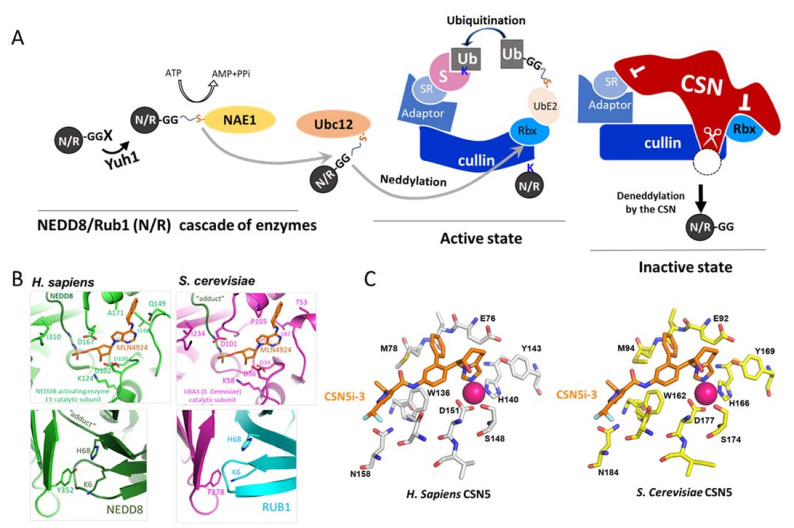
The evolutionarily conserved cullin neddylation/deneddylation enzymes. (**A**) A schematic model for cullin-RING ligase (CRL) activation and inhibition. NEDD8/Rub1 (N/R) is translated as a precursor that initially needs to be trimmed by the cysteine protease Yuh1 to expose the diglycine motif. Next, N/R is activated by ATP and forms thioester with an E1 (NAE1) enzyme. Following this step, N/R is trans-thiolized to an E2 (Ubc12) enzyme, which interacts with the RING subunit (Rbx) to transfer the N/R to a specific lysine (K) residue on a cullin. The N/R modification of cullins activate CRLs, which in return transfer ubiquitin (Ub) from Ub E2 (UbE2) enzyme to a substrate (S). Subsequently, CRLs are inhibited through N/R hydrolysis by the COP9 signalosome (CSN) complex, resulting in free N/R-GG. Notably, the CSN inhibits CRLs also non-enzymatically, through steric clashes (⊥). (**B**) Molecular model of the human (left, green) and *Saccharomyces cerevisiae* (right, magenta) NAE nucleotide binding sites with MLN4924 (top) or NEDD8/Rub1 (bottom). The *S. Cerevisiae* homology model was generated using SWISS-MODEL based on the structure of NAE1 in complex with NEDD8 and MLN4924 (PDB ID 3GZN), oriented as previously shown [25]. (**C**) Predicted active site of the *S. cerevisiae* Csn5 subunit, bound to the CSN5i-3 inhibitor. Active site of the human crystal structure of CSN5 in complex with the inhibitor CSN5i-3. Homology model was generated using SWISS-MODEL, using the PDB accession code 5JOG as a template, oriented as previously described [26]. Amino acids of the predicted active sites are shown in sticks and the zinc cations are shown in purple spheres.

**Figure 2 biomolecules-11-00497-f002:**
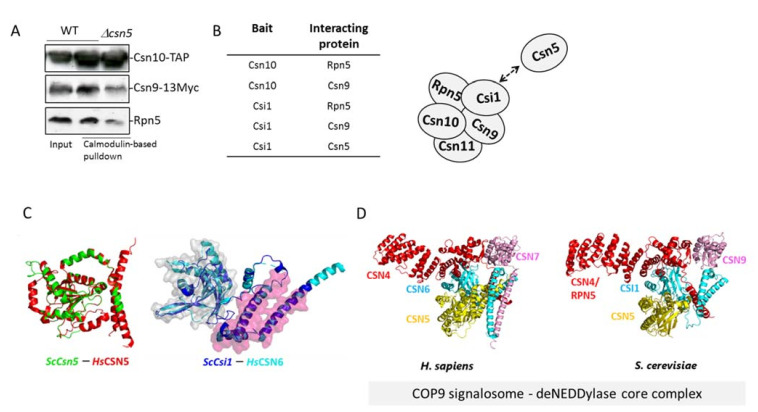
Interactions between the *S. cerevisiae* CSN subunits. (**A**) The pulldown of CSN10-TAP through calmodulin affinity resin from wildtype and *Δcsn5* mutant cells used to evaluate subunit interactions through immunoblotting for Rpn5 (by Rpn5 antibodies) and Csn9-13Myc (by Myc antibodies). (**B**) A summary of the findings in Figure 2A and Appendix A. (**C**) Superimposition of the *S. cerevisiae* Csn5 (green) and Csi1 (blue) homology models with their corresponding *H. sapiens* CSN5 and CSN6 template structures (Red and cyan respectively, PDB ID 4D10). Gray and pink surface representation is shown for the CSN6/Csi1 MPN domain (residues 35-145 in CSN6) and S6CD domain (residues 215–295 in CSN6), respectively. (**D**) Empirical 3D structure of the human and *S. cerevisiae* CSN deNEDDylase core complexes. *S. cerevisiae* homology models were built using Swiss-Model for Csn5, CSN4/Rpn5, and CSN7/Csn9, or by Phyre^2^ for Csi1, as described in Table 1 and in the Appendix A. Note that residues that are not aligned with the sequence of the template are not shown in this predicted comparative model.

**Figure 3 biomolecules-11-00497-f003:**
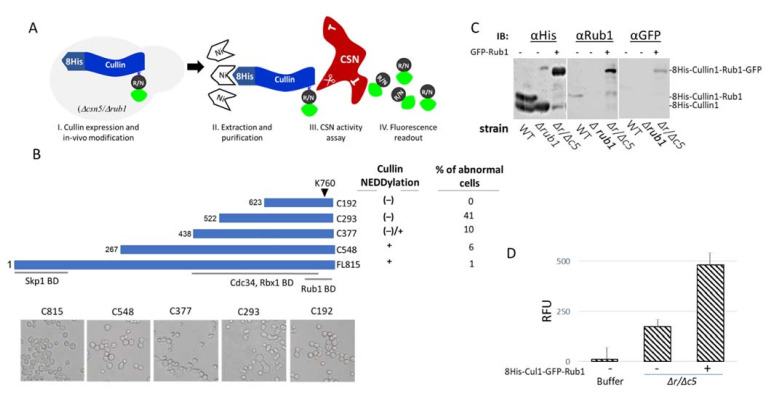
Designing a general substrate for a CSN cullin deneddylase readout assay. (**A**) Assay scheme: 8His-Cullin (yCul1) and GFP-Rub1/Nedd8 (R/N) are overexpressed in *Δcsn5Δrub1* mutant cells, and become covalently attached to each other through the endogenous neddylation cascade of enzymes. Next, the 8His-yCul1-GFP-Rub1 conjugates (e.g., substrate) are affinity purified through Ni-NTA resin. These substrates are added to CSN or CSN-containing extracts (e.g., enzyme), which eventually cause the hydrolysis of GFP-Rub1 (e.g., product) that is collected for fluorescence evaluation. (**B**) Various mutants of yCul1, all of which bear the neddylation specific residue (K760) and an N-terminal 8His tag. Terminology is according to the number of amino acids in each of the mutants. Rub1, Rbx1, and Skp1 binding domains (BD) are illustrated at the bottom. The ability of each yCul1 form to be modified by Rub1 (neddylation) is indicated by + or (−). Transformant *Δcsn5ΔRub1* mutant cells bearing the yCul1 truncations were collected directly from the agar plates and cell morphology was examined by light microscopy. Cells with abnormal morphology (enlarged or elongated) were counted, and the percentage of cells showing atypical morphology was summarized (top, right) or shown (bottom). (**C**) The ability of endogenous neddylation enzyme to modify 8His-yCul1 by GFP-Rub1 was evaluated. Wildtype (wt) *Δrub1* and *Δcsn5Δrub1 (Δr/Δ5)* expressing the fluorogenic substrate were grown in selective medium. Neddylation of the full length 8His-yCul1 (FL815) was analyzed by immunoblotting with antibodies for yCul1, 6His, and GFP. (**D**) Native protein extracts were linked to Ni-NTA resin, eluted, and fluorescent readout was detected by a plate-reader and counted in “relative fluorescent units” (RFU). Notably, cells expressing FL815 and GFP-Rub1 show high fluorescence values compared to naïve cells or buffer controls.

**Figure 4 biomolecules-11-00497-f004:**
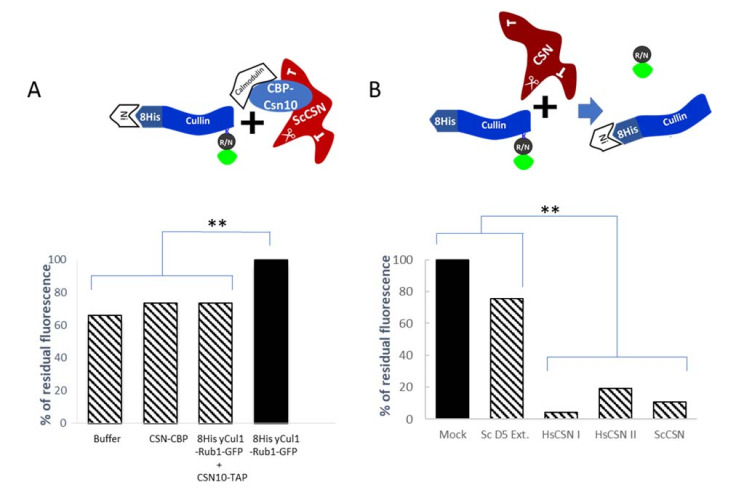
Using the *S. cerevisiae* fluorescent substrate to evaluate human CSN activity. (**A**) Ni-NTA resin (Ni)-attached substrate (8His-yCul1-GFP-Rub1) was mixed with calmodulin resin attached to CBP-Csn10 CSN complexes (*Sc*CSN) for 20 min before evaluating the fluorescent readout. Readout of the buffer, or substrate-free CSN resin were used as controls. The residual fluorescent of the substrate in the CSN-included assay (lane 3) is compared to the total fluorescence in the enzyme-free experiment (lane 4). (**B**) Crude extract of *Δcsn5Δrub1* cells expressing 8His-yCul1-Rub1-GFP exposed to purified human CSN from individual purification events (CSN I, CSN II) and to wildtype *S. cerevisiae* extract (*Sc*CSN) for 30 min at 30 °C. Following the assay, the substrate was linked to Ni-NTA resin and fluorescent readout was evaluated, compared with controls (in black): Substrate treated by buffer (mock), or with *Δcsn5* yeast extracts (Sc D5 ext.). Results were statistically verified by one-way ANOVA followed by Tukey HSD Test (*n* > 3). ** represents *p* < 0.01.

**Figure 5 biomolecules-11-00497-f005:**
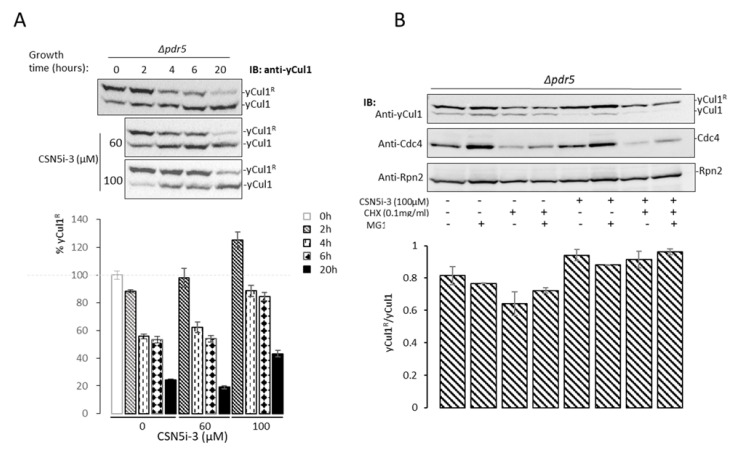
The human Csn5 inhibitor CSN5i-3 is compatible to *S. cerevisiae*. (**A**) Logarithmic *Δpdr5* cells were treated with CSN5i-3 (0, 60, and 100 µM) for indicated times (0–20 h). The neddylation status of yCul1 was calculated through IMAGEJ from three independent experiments, shown as % of neddylation from the starting 0-h time-point (bottom). (**B**) *∆pdr5* cells at the early log phase were treated with CSN5i-3 (100 µM), cycloheximide (CHX) (0.1 mg/mL), and MG132 (50 µM) to assess the inhibition effect of CSN5i-3 on deneddylation. Representative immunoblots of yCul1 show the effect of the different treatments on yCul1/yCul1^R^ ratio with the short-lived protein Cdc4 as control for CHX and MG132 inhibition treatments. Rpn2 expression is presented as a loading marker.

**Figure 6 biomolecules-11-00497-f006:**
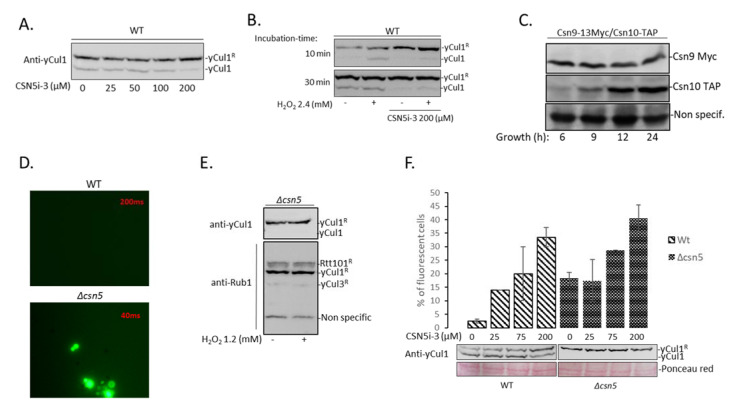
CSN activity in oxidized condition. (**A**) Logarithmic wildtype (WT) cells were treated with CSN5i-3 for 2 h as indicated and immunoblotted for yCul1. (**B**) Logarithmic wildtype cells were treated with 200 µM of CSN5i-3 for 2 h before adding 2.4 mM H_2_O_2_ for 10 or 30 min before immunoblotted for yCul1. (**C**) The expression of CSN subunits was evaluated overtime in a wildtype strain in which Csn9 is C-terminally tagged by 13Myc and Csn10 is C-terminally TAP tagged. Myc antibodies and rabbit IgG were used for immunoblotting. (**D**) Wildtype and *Δcsn5* cells were treated with DCFDA and fluorescence was viewed by microscope. Notably, the mutant fluorescence was much higher than wildtype; thus, images were taken in a lower intensity (40 ms). (**E**) Logarithmic *Δcsn5* mutant cells were treated with H_2_O_2_ for at least 30 min and cullin neddylation status was assessed by immunoblotting with antibodies for yCul1 (top) or for Rub1 (bottom), recognizing the three yeast neddylated cullins (yCul1, Rtt101, and yCul3). (**F**) Endogenous ROS levels of wildtype and *∆csn5* cells with and without the inhibitor (0, 25, 75, and 200 µM) were observed following DCFDA treatment. Two random fields were selected for each sample during quantification. The average intensity was calculated based on the number of fluorescent cells in the fields. Immunoblots are shown for yCul1 neddylation status; ponceau red is a loading marker.

**Table 1 biomolecules-11-00497-t001:** Sequence identity between the *H. sapiens* and *S. cerevisiae* orthologues CSN subunits.

H. sapiens vs. S. cerevisiae	CSN1 vs. Csn11 [*1]	CSN4 vs. Rpn5 [*2]	CSN5 vs. Csn5 [*3]	CSN6 vs. Csi1 [*4]	CSN7 vs. Csn9 [*5]	CSN2 vs. Csn10 [*6]
Sequence identity (%)	16%	100% to the 19S lid Rpn5	39.3%	15%	23.4%	20%
Coverage (%)	85%	100%	47%	82%	77%	84%
GMQE	0.38	0.76	0.25	0.5	0.39	0.12
QMEAN	−5.57	−2.61	−0.62	−4.08	−1.2	−4.7
** Confidence (%)	99.7%			98.0%		100%

[*1] Built on RPN7 crystal structure (PDB ID:4CR2, [56]). [*2] CSN4-Rpn5 is identical to the proteasomal Rpn5 subunit. Based on the Rpn5 crystal structure (PDB ID:5MPD, [53]). [*3] Built on CSN5 crystal structure (PDB ID:4D18, [37]). [*4] Built on CSN6 crystal structure (PDB ID:4D10, [37]). [*5] Built on *A. thaliana* CSN7 crystal structure (PDB ID: 3CHM, [54]). [*6] Built on CSN2 crystal structure (PDB ID:6R7N, [57]). Global model quality estimation (GMQE) > 0 is an estimator that combines properties from the target–template alignment and the template structure [58]. The qualitative model energy analysis (QMEAN) < (−4) provides estimation for the “degree of nativeness” of the structural features in the model [59]. ** Confidence values were calculated for Csn11, Csi1, and Csn10 through Phyre^2^ [60]. Confidence represents the probability (0–100%) for true homology between the aligned sequences and their templates (see “Appendix A” for additional information).

## Data Availability

Not applicable.

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
