# Peer review of "Saccharomyces cerevisiae as a Toolkit for COP9 Signalosome Research"

_biomolecules, 2021, doi:10.3390/biom11040497_

Round 1
Reviewer 1 Report
The COP9 signalosome (CSN) is a multiple-subunit protein that is conserved across species. The role of CSN includes substrate ubiquitination, degradation, and regulating cullin RING ligase activities. Here in this paper, CSN deneddylase core complex was found to be conserved in S. cerevisiae. The sensitive fluorescence readout assay was developed for biochemical quantification of cullin deneddylation by CSNs from various origins. Csn5 was shown to be targeted by the inhibitor targeting the human orthologue. The human inhibitor lead to the accumulation of neddylated cullins and the formation of reactive oxygen species. The paper provided a way to use S. cerevisiae to study CSN deneddylation and inhibitor testing. The paper has many interesting experimental designs.
Major comment:
- In the introduction section (lines 94-95), please introduce more about the approaches that the authors used.
- The authors should explain what are the advantages the truncated mutants can have in figure 3B.
- Line 241: Please explain how does figure 2C reflects the MPN structural fold of Csi1?
- Line 422: CSN5i-3 treatment slows down the oxidation, but it further found to induce ROS? Please clarify the discrepancy.
Minor points:
- Line 52: “(“ is missing.
- Please rewrite the sentence in lines 57-59.
- Line 216: change “ too,” to “also”.
- Please rewrite the sentence in lines 220-222.
- Please reword the sentence in line 242-243.
- The full name of RFU should also be spelled out in the legend of figure 3D.
- Please rephrase the sentence in line 350-351.
- “then” should be “than” in line 291.
- Please enlarge the text in figure 3A.
- In figure 4A, please spell out the full name Cal as calmodulin.
- Label the Y-axis in the graph of Figure 4.
- Line 313: delete “of”
- Line 352: Delete one “ Accordingly”
- Line 383: “60 M”
- Line 442: “his” should be “this”
Author Response
REVIEWER 1
We would like to thank the reviewer for the valuable comments. We changed the text according to the comments, and added a figure (6E).
Major comment:
- In the introduction section (lines 94-95), please introduce more about the approaches that the authors used.
Added, please see lines 96-99: “In this manuscript, we describe a sensitive fluorescence readout assay, suitable for biochemical assessment of cullin deneddylation by CSNs from distinct organisms, such as H. sapiens and S. cerevisiae. We also exhibit that the S. cerevisiae Csn5 is inhibited by CSN5i-3, an inhibitor that targets the human orthologue. “
- The authors should explain what are the advantages the truncated mutants can have in figure 3B.
We thank the reviewer for this comment. We initially thought that the truncations will allow selecting the best CSN substrate (lines 273-274): “Since yCul1 is a large protein, essential for yeast viability, we wanted to select a protein size with a minimal effect on vitality”. But this was not the case. The advantages eventually found (lines 316-319): “Although the truncation mutants did not provide a practical advantage to the assay, they can still be useful, especially given the dominant negative activity of C293. This mutant lacks the N terminal arm of yCul1 that is involved in the selection of SCF substrates, thus, could be suggested as an original way to identify additional SCF substrates that have not yet been identified”.
- Line 241: Please explain how does figure 2C reflects the MPN structural fold of Csi1?
We are sorry for not being clear. We corrected the figure legend (lines 253-254): “Grey and pink surfaces representation is shown for the CSN6/Csi1 MPN domain (residues 35-145 in CSN6) and S6CD domain (residues 215-295 in CSN6), respectively”. We also corrected lines 248-251: “Interestingly, molecular modeling of Csi1 on available structures of Csn6 suggest an MPN structural fold for Csi1, as seen from the superimposition of the N terminal areas of the S cerevisiae Csi1 (dark blue) and the human Csn6 (light blue) (Figure 2 C, grey surface)”.
- Line 422: CSN5i-3 treatment slows down the oxidation, but it further found to induce ROS? Please clarify the discrepancy.
We thank the reviewer for this important and critical comment. Indeed, we suspect that CSN5-i3 induces ROS in S. cerevisiae, regardless to the inhibition of cullin deNEDDylation. As we at the end of the manuscript: “Our results suggest that high doses of the inhibitor affect additional physiological path-ways in yeast, which needed to be further deciphered. The data could only be obtained because of the non-essentiality of Csn5 in this organism”. However, according to this comment we decided to further discuss and clear this point. ROS leads to low yCul1 neddylation status (Bramasole et al. Redox Biology 2019), and CSN5i-3 plays two opposing functions: promotes ROS that inhibits the neddylation cascade and inhibits the CSN deneddylase activity. Despite this, we see accumulation of yCul1-R, suggesting that one force (inhibition of deneddylation) is stronger than the other force. In line with this idea, yCul1 is fully neddylated in ∆csn5 cells that accumulate endogenous ROS (Figure 5D). We added a new figure (Figure 6E) to show that the treatment of ∆csn5 with H2O2 does not affect the cullins neddylation status. We added explanation to the text (lines 450-460): “Indeed, treatment of ∆csn5 cells with DCFDA, revealed that Csn5-deficient cells produce high ROS (Figure 6D). The findings that ∆csn5 mutant cells produce high ROS may be in conflict with the overall total accumulation of neddylated yCul1 in this mutant [45], since ROS inhibits the neddylation cascade [48], and therefore a population of non-neddylated cullins should exist as well. Despite this estimation, even treatment of ∆csn5 mutant cells with H2O2 revealed total accumulation of neddylated yCul1, yCul3 and Rtt101, the three S. cerevisiae cullins (Figure 6E). The results are in line with the finding that treatment of wildtype cells with CSN5i-3 lead to a high yCul1 neddylation status, even though the cells produce ROS (Figure 6F). These findings can be explained if single-enzyme removal of Rub1 is a more efficient process than the reverse process that requires a series of enzymes”.
Minor points: * Note that the original location of the comments has changed, therefore the corrections are elsewhere.
- Line 52: “(“ is missing. Corrected
- Please rewrite the sentence in lines 57-59. Corrected
- Line 216: change “ too,” to “also”. Corrected
- Please rewrite the sentence in lines 220-222. Corrected
- Please reword the sentence in line 242-243. Corrected
- The full name of RFU should also be spelled out in the legend of figure 3D. Corrected
- Please rephrase the sentence in line 350-351. Corrected
- “then” should be “than” in line 291. Corrected
- Please enlarge the text in figure 3A. Corrected
- In figure 4A, please spell out the full name Cal as calmodulin. Corrected
- Label the Y-axis in the graph of Figure 4. Corrected
- Line 313: delete “of” Corrected
- Line 352: Delete one “Accordingly” Corrected
- Line 383: “60 mM” deleted
- Line 442: “his” should be “this” Corrected
Reviewer 2 Report
The manuscript titled “Saccharomyces Cerevisiae as a Toolkit for COP9 Signalosome Research” described a study on S. cerevisiae CSN. The authors developed a sensitive fluorescence readout assay, suitable for biochemical quantification of cullin deneddylation by CSNs from various origins. They also showed the human CSN5 inhibitor CSN5i-3 can also inhibit S. cerevisiae Csn5, leading to the accumulation of neddylated cullins and the formation of reactive oxygen species.
However, there are several issues in the manuscript. Thus, I suggest the authors to revise their manuscript.
Detailed comments are listed below.
Major issues:
1) Figure 1, Table 1 and Figure 2, the authors didn’t show the model quality estimation results of their generated models by Swiss-Model. They should provide the model quality estimation results as well as the detailed sequence alignments of each protein with their template sequence. If the model quality is low, the authors should be careful to draw any conclusion from these modelled structures.
2) Figure 3D and Figure 4, the bar diagrams didn’t have any error bars. Please add error bars based on the results of triplicate experiments.
3) Figure 4, the protein level of CSN in all kinds of “extracts” were not quantified. As a result, unless demonstrated otherwise by the authors, this toolkit can only be used as a qualitative assay, which has limited applications.
4) Figure 5A, “0” hour results should be included in the gel.
5) Figure 5B, after treatment of CHX and MG132 (Lane 4 and Lane 8), the protein levels of yCul+yCulR and even the input control of Cdc4 were not the same. Lane 4 had more proteins than Lane 8; Figure 6B, the protein levels of yCul+yCulR were quite different among lanes. All the immunoblotting experiments in this study should be performed with loading controls.
Other minor issues:
1) Line 51, PCI (proteasome, CSN, eIF3) à PCI (proteasome lid, CSN, eIF3)
2) Line 52, the left parenthesis for MPN full name is missing.
3) Line 54-55, deubiquitylation à deubiquitination
4) Line 73-76, please indicate the species of the mentioned CSN because residue 104 of CSN5 was mentioned.
5) Line 25, 76, 78, 119, 392, Csn5i-3 à CSN5i-3
6) Line 76-79, reference for CSN5i-3 should be added.
7) Line 80, stearic à steric
8) Line 82, require à which are required
9) Line 85, CSN complex-free formà CSN complex Csn5-free form
10)Line 141, please add “nm” for wavelength
11) Line 153, add “,” between 2’ 7’
12) Line 163, please add “nm” for wavelength
13) Line 175, stearic à steric
14) Line 187, pevonedistat à Pevonedistat
15) Table 1, H. Sapiens à H. sapiens
16) Figure 3C, label each band correctly, instead of “8His”
17) Figure 3D, Δn à Δr
18) Line 289, 2-micron of what?
19) Line 291, then à than
20) Line 313, expression of what?
21) Line 342, fluoresce à fluorescence
22) Line 352, delete “Accordingly,”
23) Figure 5A, “4h” bar fill is different color from indicated fill.
24) Line 376, please correct the concentration of CHX
25) Line 396, CSN5i-3CSN à CSN5i-3
26) Line 442, his information??
Author Response
We would like to thank the reviewer for the valuable comments. We corrected the text, added SD+/- to figure 3D, replaced the required blots in figures 5a and 5b and added a supplemental alignment part.
Major issues:
1) Figure 1, Table 1 and Figure 2,
- the authors didn’t show the model quality estimation results of their generated models by Swiss-Model.
We thank the reviewer for the comment. We now used the GMQE and QMEAN parameters to determine and to ascertain the model’s quality. GMQE (Global Model Quality Estimation) combines properties from the target–template alignment and the template structure (Biasini et al., 2014). The GMQE score is expressed as a 0 to 1 (high number indicates higher reliability). The QMEAN Z-score is expressed as numbers between (-4) to 0 and provides estimation for the "degree of nativeness" of the structural features in the model (Benkert et al., 2011).
They should provide the model quality estimation results as well as the detailed sequence alignments of each protein with their template sequence.
The parameters were added to the tables (Table 1, Table S4, Figure S1B). We also added at the end of the supplemental material a “Supplemental Alignment” part that includes the parameters and alignments as requested.
If the model quality is low, the authors should be careful to draw any conclusion from these modelled structures.
In three cases the QMEAN parameters were low (for Csi1, Csn10 and Csn11). The information is detailed at the supplemental data (pages 14, 17, 18) and described in the text (lines 248-258).
2) Figure 3D and Figure 4, the bar diagrams didn’t have any error bars. Please add error bars based on the results of triplicate experiments.
We are sorry that the error bars for figure 3D had lost in the previous version. It is now corrected. Error bars were not added to figure 4A. The various experiments revealed a similar pattern, but different ratios. We claim that the experiment is not good due to inconsistency and high background. Figure 4B represents 3 independent repeats instead of averaging them.
3) Figure 4, the protein level of CSN in all kinds of “extracts” were not quantified. As a result, unless demonstrated otherwise by the authors, this toolkit can only be used as a qualitative assay, which has limited applications.
We thank the reviewer for the comment. This issue is indeed addressed in the conclusions: “Future studies will include adjustment of the assay for detection of endogenous levels of CSN activity which could further serve as an inexpensive universal method to assess CSN activity not only in model organisms, but in non-model organisms in their natural habitats”. To be more precise, the word "quantification" in the abstract and in the introduction has been replaced by "assessment."
4) Figure 5A, “0” hour results should be included in the gel.
We thank the reviewer very much for this important comment. Due to this comment, we realized that we were wrong in the chart, by adding an 8h timepoint (which we hadn’t took) and marked the zero time as 2 hours. The data is now corrected as well as this figure legend.
5) Figure 5B, after treatment of CHX and MG132 (Lane 4 and Lane 8), the protein levels of yCul+yCulR and even the input control of Cdc4 were not the same. Lane 4 had more proteins than Lane 8; Figure 6B, the protein levels of yCul+yCulR were quite different among lanes. All the immunoblotting experiments in this study should be performed with loading controls.
We thank the reviewer for this comment. We re-blotted the Cdc53 membrane with a first antibody for Cdc4. Cdc4 is not a loading control. It is a short-lived proteasome substrate, which we used to confirm the inhibition of proteasome through MG132 (stabilization of Cdc4), or the inhibition of ribosome through CHX (fast turnover of Cdc4). According to the comment, we blotted the membrane by Rpn2 antibody, which we used as a loading marker (see below). According to Rpn2, the loading in lanes 3 and 7 (CHX +/- CSN5i-3) are similar, and as expected, the turnover of Cdc4 is faster than in the untreated samples (lanes 1 and 5). As expected, the loading marker, Rpn2 does not show major differences between lanes 4 and 8 (the combination of CHX and MG132 with or without CSN5i-3), while the turnover of Cdc4 is compensated. By comparing Cdc53/yCul1 to the expression patterns of Rpn2 and Cdc4, we were able to confirm that the loss of the unmodified yCul1 is not due to changes in its expression/degradation pattern, but through the inhibition of CSN activity (by CSN5i-3). We replaced the figure and explained it better in the text. The original blot that was used for IB is shown in the enclosed document.
Other minor issues:* Note that the original location of the comments has changed, therefore the corrections are elsewhere.
1) Line 51, PCI (proteasome, CSN, eIF3) à PCI (proteasome lid, CSN, eIF3). Corrected
2) Line 52, the left parenthesis for MPN full name is missing. Corrected
3) Line 54-55, deubiquitylation à deubiquitination Corrected
4) Line 73-76, please indicate the species of the mentioned CSN because residue 104 of CSN5 was mentioned. Added “In human, ….”
5) Line 25, 76, 78, 119, 392, Csn5i-3 à CSN5i-3. Corrected
6) Line 76-79, reference for CSN5i-3 should be added. Added
7) Line 80, stearic à steric. Corrected
8) Line 82, require à which are required. Corrected
9) Line 85, CSN complex-free form à CSN complex Csn5-free form – Corrected
“10) Line 141, please add “nm” for wavelength. Corrected all over the text.
11) Line 153, add “,” between 2’ 7’ Corrected
12) Line 163, please add “nm” for wavelength Corrected
13) Line 175, stearic à steric Corrected
14) Line 187, pevonedistat à Pevonedistat Corrected
15) Table 1, H. Sapiens à H. sapiens Corrected
16) Figure 3C, label each band correctly, instead of “8His” Thank you for the comment. The caption in the previous figure has been cut and is now replaced
17) Figure 3D, Δn à Δr Corrected
18) Line 289, 2-micron of what? corrected to “expressed in a multi-copy plasmid”
19) Line 291, then à than Corrected
20) Line 313, expression of what? Corrected to “expression of the truncated mutants”
21) Line 342, fluoresce à fluorescence Corrected
22) Line 352, delete “Accordingly,” Corrected
23) Figure 5A, “4h” bar fill is different color from indicated fill. Corrected
24) Line 376, please correct the concentration of CHX Corrected to 0.1 mg/ml
25) Line 396, CSN5i-3CSN à CSN5i-3 Corrected
26) Line 442, his information?? Corrected to “This information …”.

Round 2
Reviewer 2 Report
The authors have addressed most of the issues raised by the reviewers previously.
There are still some issues:
Major issue:
As the authors presented, the modeling results by Swiss-Model are quite poor for Csi1, Csn10 and Csn11, which is expected because Swiss-Model highly relies on the correct template provided for modelling. However, currently there are no good templates for Csi1, Csn10 and Csn11.
Line 264-265, since the author used human CSN6 as template for Csi1 modeling using Swiss-Model, it’s not surprised to me that the Csi1 model also has MPN fold at N-terminal, and Csi1 model superimposed very nicely with human CSN6 structure. The author should be extremely careful to draw any conclusion from these poorly modelled structures.
I would recommend the authors to use other better modeling tools: trRosetta, ITASSER, RaptorX, tfold, etc.
For example, I ran the structure prediction of Csi1 using ITASSER, and got the closest model for Csi1 is PDB: 5a5t chain F, which is eif3F subunit.
I highly recommend the authors to re-run structure predictions, and they may get better and more reliable models to further interpret their functions.
Minor issues:
1) Figure 3C, Δn à Δr
2) Line 276-277, why are these texts highlighted?
Author Response
Major issue:
As the authors presented, the modeling results by Swiss-Model are quite poor for Csi1, Csn10 and Csn11, which is expected because Swiss-Model highly relies on the correct template provided for modelling. However, currently there are no good templates for Csi1, Csn10 and Csn11.
We thank the reviewer to the comment. According to this comment "Confidence values" were calculated for Csn11, Csi1 and Csn10 through Phyre2. Confidence represent the probability (0 - 100%) for true homology between the aligned sequences and their templates. We also added the calculations to the supplementary materials.
Line 264-265, since the author used human CSN6 as template for Csi1 modeling using Swiss-Model, it’s not surprised to me that the Csi1 model also has MPN fold at N-terminal, and Csi1 model superimposed very nicely with human CSN6 structure. The author should be extremely careful to draw any conclusion from these poorly modelled structures.
We thank the reviewer. Indeed, as we wrote above, Phyre2 was calculated. we also changed the text accordingly "Interestingly, molecular modeling of Csi1 on available structures of the human orthologue of CSN6 by Swiss-Model, suggested an N-terminal MPN structural fold. Since Swiss-Model is highly relying on the correct template provided for modelling, we ran structure prediction on Csi1 using Phyre2. Interestingly, using this modelling tool, the closest model for Csi1 was found as PDB ID 5A5T of the human eIF3F (98.0% confidence, 13% sequence identity), one of the two MPN subunits of the eIF3, which is in fact a paralogue of CSN6/Csi1. CSN6 appeared second on the Phyre2 list, with 97.5% confidence and 15% sequence identity. We used CSN6 (Figure 2 C) and eIF3F (Figure S3 A) as templates to predict Csi1 structure. "
I would recommend the authors to use other better modeling tools: trRosetta, ITASSER, RaptorX, tfold, etc. For example, I ran the structure prediction of Csi1 using ITASSER, and got the closest model for Csi1 is PDB: 5a5t chain F, which is eif3F subunit.
Indeed, we received similar results through Phyre2.
I highly recommend the authors to re-run structure predictions, and they may get better and more reliable models to further interpret their functions.
Thank you very much for this valuable comment!!
Minor issues:
1) Figure 3C, Δn à Δr - CORRECTED
2) Line 276-277, why are these texts highlighted? - CORRECTED